# Differential Response of Leafminer Flies *Liriomyza trifolii* (Burgess) and *Liriomyza sativae* (Blanchard) to Rapid Cold Hardening

**DOI:** 10.3390/insects12111041

**Published:** 2021-11-19

**Authors:** Junaid Iqbal, Xiao-Xiang Zhang, Ya-Wen Chang, Yu-Zhou Du

**Affiliations:** 1College of Horticulture and Plant Protection & Institute of Applied Entomology, Yangzhou University, Yangzhou 225009, China; junaidagri1@gmail.com (J.I.); zxxyzu@yahoo.com (X.-X.Z.); changyawen@yzu.edu.cn (Y.-W.C.); 2Joint International Research Laboratory of Agriculture and Agri-Product Safety, The Ministry of Education, Yangzhou University, Yangzhou 225009, China

**Keywords:** *Liriomyza* spp., sudden temperature change, rapid cold hardening, cold tolerance, supercooling point

## Abstract

**Simple Summary:**

*Liriomyza trifolii* (Burgess) and *L. sativae* (Blanchard) are closely-related, polyphagous leafminers that occur worldwide and presumably compete with each other. In this study, we evaluated the response of pupae and adults from both species to acute (2 h) cold exposures. The results were used to identify the lethal temperature for 80% of the population (LT80) for each species. In a separate set of experiments, insects were cooled to one of six nonlethal temperatures (0–5 °C) for 4 h and then cooled to the LT80 for 2 h to evaluate their rapid cold hardening (RCH) response. *L. trifolii* exhibited stronger cold tolerance than *L. sativae*; furthermore, the supercooling point of *L. trifolii* was significantly lower than that of *L. sativae*. RCH was induced in pupae of both species at a range of low temperatures (0–5 °C), and *L. sativae* pupae showed a more robust RCH response (e.g., lower supercooling pointand more durable RCH) than *L. trifolii* pupae. Our results indicate that subtle differences in RCH and basal cold tolerance impact the competitiveness of the two leafminers.

**Abstract:**

Rapid cold hardening (RCH) is a rapid and critical adaption of insects to sudden temperature changes but is often overlooked or underestimated as a component of survival. Thus, interspecific comparisons of RCH are needed to predict how phenotypes will adapt to temperature variability. RCH not only enhances cold survival but also protects against non-lethal cold injury by preserving essential functions such as locomotion, reproduction, and energy balance. This study investigated the difference in basal cold tolerance and RCH capacity of *L*. *trifolii* and *L*. *sativae*. In both species, the cold tolerance of pupae was significantly enhanced after short-term exposure to moderately cold temperatures. The effect of RCH last for 4 h in *L. sativae* but only 2 h in *L. trifolii*. Interestingly, *L. trifolii* adults had a RCH response but *L. sativae* adults failed to acclimate. Short-term acclimation also lowered the supercooling point significantly in the pupae of both species. Based on these results, we propose a hypothesis that these differences will eventually affect their competition in the context of climate change. This study also provides the basis for future metabolomic and transcriptomic studies that may ultimately uncover the underlying mechanisms of RCH and interspecific competition between *L*. *trifolii* and *L*. *sativae*.

## 1. Introduction

*Liriomyza trifolii* (Burgess) and *L. sativae* (Blanchard) (Diptera: Agromyzidae) are polyphagous leafminers that feed on a variety of horticultural crops. The two species are closely related with similar genitlalic morphologies [1], life histories, and host ranges [2,3]. They are native to the Americas but have rapidly spread worldwide [4]. *L. trifolii* and *L. sativae* are among the most significant leafmining pests in China [5,6,7]. After its introduction to Hainan Province in October 1993, *L. sativae* spread throughout China within a few years [8,9]. *L. trifolii* invaded China after *L. sativae*. This species was first recorded in Guangdong Province in 2005, after which it rapidly extended to another 11 provinces where it caused significant damage to a variety of horticultural plants [10]. Since the introduction of *L. trifolii*, it has displaced *L. sativae* and become the dominant species in some regions of southern China [6,11]. *L. trifolli* has proven to be a more damaging pest of crops in the regions of China that it has invaded [12,13]. Due to its increased cold tolerance and lower supercooling point (SCP), *L. trifolii* has the potential to displace *L. sativae* and expand its range to northern China [14]. Climatic stress, overwintering, and cold tolerance are temperature-related factors that determine the geographical distribution of leafminers [5,15].

Temperature affects both the distribution and abundance of insects [16,17]. Interactions between *Liriomyza* spp. are temperature-dependent and slight differences in temperature tolerance can shift the competitive balance between species [18,19,20]. Cold tolerance is one of the most important factors affecting the survival and distribution of insects [21]. The quantification of cold tolerance, including determining the SCP, can be fundamental to evaluating low temperature performance in insects and the pervasive effects of temperature [22]. The SCP is defined as the temperature where body fluids spontaneously freeze and is a commonly used index for evaluating cold tolerance in insects [23]. Interactions between *L. sativae* and *L. trifolii* are complex and are likely impacted by additional factors, including fecundity, natural enemies, and interspecific hybridization [24,25].

Owing to global warming, fluctuations in climate have increased and are predicted to continue. Due to abnormal temperature fluctuations during early spring and late autumn, rapid cold hardening (RCH) plays an important role in insect survival. Therefore, studies of cold tolerance have become more prominent in insect ecology, physiology and genetics [16]. Cold tolerance can be achieved through long term cold acclimatization, in which insects undergo natural and gradual temperature changes while through rapid cold hardening (RCH) insects experience a sudden decrease in the temperature for a short time [26]. There are considerable mechanistic differences between acclimation and RCH (see review by Teets et al.) [27]. In autumn and spring months, overwintering insects are often exposed to sudden decreases in temperature for short periods [28,29]. In lab experiments, insects pre-treated with a nonlethal low temperature for a short period of time acquired the ability to survive subsequent low temperatures that would otherwise be lethal, which is an indicator of RCH [29]. RCH can increase cold survival in short durations of time (e.g., 30 min), whereas seasonal acclimation generally requires a much longer period of time to enhance cold tolerance [30,31]. RCH has been documented in a variety of insects and contributes to survival at temperatures that were previously lethal [29,32]. In addition to increasing survival, RCH preserves essential ecological functions needed to survive sublethal stress [27]. A combination of acclimation and RCH contribute to increased stress resistance in the field [33].

The present study was undertaken to identify differences in cold tolerance between *L. sativae* and *L. trifolii* and the effects of RCH on the survival of both species. We investigated the SCP and RCH of *L. sativae* and *L. trifolii* to better understand interspecies competition. Here, we address a hypothesis that RCH will affect the interspecific competition between *L. sativae* and *L. trifolii* and eventually the replacement of *L. sativae* by *L. trifolii* in northern regions of China will take longer than predicted by previous studies [14,19]. This research also provides a baseline for further metabolomic, proteomic, and transcriptomic studies of RCH, which will help elucidate the mechanistic basis of the phenomenon response and its role in interspecific competition.

## 2. Materials and Methods

### 2.1. Insect Rearing

Leaves infested with leafminers were collected from greenhouses in Yangzhou (32.39° N, 119.42° E), China in 2020. Based on their morphological characteristics, adults were identified as *L. trifolii* or *L. sativae*. Both species were reared on kidney beans at 26 °C with a 16:8 light/dark photoperiod as described by Chen and Kang [34]. Bean seeds (4–5/pot) were sown and cultivated until 4–5 true leaves were present; bean seedlings were then transferred into cages (40 cm × 40 cm × 65 cm) for leafminer feeding. Leaves with tunnels were collected and placed in plastic bags for pupation and the collected pupae from these bags were transferred into glass tubes. No wild populations from the field were mixed into the lab population during the experimental period.

### 2.2. Critical Temperature

To evaluate the low-temperature survival of leafminers and the effects of RCH, experiments were conducted to determine the critical temperature causing 80% mortality (lethal temperature 80; LT80) [29,35]. Pupae (2, 5, and 7 d) and adults were subjected to the following subzero temperatures: −7, −8, −9, −10, −11 and −12 °C. Pupae and adults were collected in small glass tubes, transferred into a temperature controller (DC-3010, Ningbo Jiangnan instrument factory, Ningbo, China), and exposed to the subzero temperature for 2 h. After treatment, pupae were transferred to a controlled environmental chamber for emergence; the survival of adults was noted after 20 min of recovery. Adults who showed no movement when touched with a brush were recorded as dead. Experiments consisted of three replicates of 30 individuals; each replicate and control (30 individuals) was tested. Survival and corrected percentage survival were calculated by the following formulas:$$\mathrm{Survival}\;=\frac{\mathrm{No}.\;\mathrm{of}\;\mathrm{adults}\;\mathrm{emerged}/\mathrm{alive}\;}{\mathrm{Total}\;\mathrm{no}.\;\mathrm{of}\;\mathrm{pupae}/\mathrm{adults}}\times 100\;\mathrm{Corrected}\;\mathrm{percentage}\;\mathrm{survival}\;=\frac{\mathrm{survival}\;\mathrm{of}\;\mathrm{treatment}\;}{\mathrm{survival}\;\mathrm{of}\;\mathrm{control}}\times 100$$

### 2.3. Assessment of the Rapid Cold Hardening (RCH) Response

To determine whether pupae and adults of *Liriomyza* spp. have the capacity to be rapidly cold hardened, pupae of different ages (2, 5, and 7 d) and adults (2 d) were chilled at sublethal temperatures ranging from 0 to 5 °C for 4 h prior to exposure to the critical temperature (LT80) for 2 h. Adult survival was recorded as noted above; pupae were transferred to an environmental chamber and monitored for adult emergence to assess survival. The purpose of treating pupae of different ages with different hardening temperatures was to identify the best combination of the two (enabling highest survival), for further experiments (durability of RCH and SCP). The temperatures enabling the highest survival rates were used as the acclimation temperature for both species. Three replications for each temperature (30 individuals/replication) were tested.

To determine the duration of the RCH response, pupae treated with the optimal hardening temperature for 4 h were transferred to 26 °C and maintained for 0, 0.5, 1, 2, and 4 h prior to exposure to the critical temperature for 2 h; after which survival was checked.

### 2.4. Determination of Supercooling Point (SCP) 

First, the supercooling point (SCP) was determined in both species in pupae of different ages, which served as a control group. These included 2-, 5-, and 7-day-old pupae. For RCH groups, age of the pupae and the hardening temperature giving highest survival in both species were selected. Control groups were directly taken from 26 °C while for the RCH groups pupae were first exposed to the best hardening temperature for 4 h then supercooling points were recorded directly after 4 h exposure. SCPs were measured using a thermocouple connected to a digital temperature recorder (UT-325, Uni-Trend technology, Dongguan, China). The tip of the thermocouple was inserted into the lid of an Eppendorf tube and pupae were attached to thermocouples by coating the tip with a thin film of petroleum jelly. Tubes containing the thermocouples and attached pupae were exposed to cold temperatures as described by Chen and Kang [34]. The lowest temperature at which an abrupt increase in temperature occurred was recorded as the SCP. 

### 2.5. Statistical Analysis

The survival of pupae and adults exposed to low temperatures was analyzed by a nonlinear regression model using GraphPad Prism v. 7.0 for the determination of the critical temperature. One-way ANOVA (followed by Tukey’s multiple comparison) was used to detect significant differences between different RCH treatments on survival and SCP using SPSS v. 16.0 (SPSS, Chicago, IL, USA). For ANOVA, data were transformed for homogeneity of variances (*p* < 0.05). A two-way ANOVA was used to detect the RCH response where species and temperature were explanatory variables and survival was the dependent variable. Similarly, for durability of the RCH, species and duration were explanatory variables. The SCPs of RCH and no RCH groups were compared to check for differences. The student’s t-test was used to compare differences in RCH and no RCH groups with SPSS v. 16.0; differences were considered significant at *p* < 0.05.

## 3. Results

### 3.1. Critical Temperature

The mean survival of 2-day-old pupae is shown for both species in Figure 1. Survival of the pupae was significantly reduced with each 1 °C reduction in temperature. Two-day old pupae of *L. trifolii* and *L. sativae* have similar survival rates after exposure to −7 °C, which were 73.3 and 71.43%, respectively (Figure 1A). However, differences in survival of the two species became more apparent when they were exposed to further cold stress (−8, −9, −10, −11 and −12 °C). The critical temperature for 2-day-old pupae that resulted in approximately 80% mortality following the 2 h exposure period was −10.4 °C for *L. trifolii*, which is nearly 2 °C lower than *L. sativae* (−8.4 °C). 5-day-old pupae (Figure 1B) and 7-day-old pupae (Figure 1C) of both species showed similar survival tendencies to 2-day-old pupae. The critical temperature of 5-day-old pupae of *L. trifolii* and *L. sativae* were −11.6 °C and −8.9 °C, respectively. Critical temperatures for 7-day-old pupae of *L. trifolii* and *L. sativae* were −10.6 °C and −8.4 °C, respectively. In both species, 5-day-old pupae showed high cold tolerance as compared to 2- and 7-day-old pupae. In the case of the adults, a similar survival pattern was found to the pupae. Mean survival data of adults are shown in Figure 1D. The results revealed that by lowering the temperature, survival also became significantly lower. Critical temperatures for adults of *L. trifolii* and *L. sativae* were similar, at −10.5 and −10.4 °C, respectively.

### 3.2. Assessment of RCH and Its Response to Different Hardening Temperatures

The mean survival of pupae and adults exposed to cold stress after cold hardening (0 to 5 °C for 4 h) is shown in Figure 2. The survival of both species at all critical temperatures was dramatically increased from 20% to over 40% after a 4 h acclimation period. Significant differences were recorded between the critical temperature and different hardening temperatures of 2-day-old pupae of both species (*L. trifolii*: *F*_6,15_ = 38.62, *p* < 0.001, *L. sativae*: *F*_6,15_ = 14.62, *p* < 0.001). Further two-way analysis of survival of 2-day-old pupae of both species compared with hardening temperatures showed no significant differences (*F*_6,42_ = 2.141, *p* = 0.08) (Figure 2A). The highest mean survival rate of 5-day-old *L. sativae* pupae was recorded when cold hardening occurred at 1 °C (91%) followed by 5 °C (88%); survival was lowest at 0 °C (76%) (*F*_6,14_ = 91.5, *p* < 0.001) (Figure 2B). As for *L. trifolii*, survival at 5 °C was significantly higher than that at 4 °C (*F*_6,14_ = 25.3, *p* < 0.001). It was found that interaction between hardening temperatures and 5-day-old pupae (both species) showed a significant difference in their survival rates (*F*_6,42_ = 8.5, *p* < 0.001). Survival of 7-day-old pupae of both species exhibited the highest survival rates after acclimation at 1 °C for 4 h (Figure 2C) (*L. sativae*: *F*_6,14_ = 35.7, *p* < 0.001; *L. trifolii*: *F*_6,14_ = 34.5, *p* < 0.001). Furthermore, interaction between 7-day-old pupae (both species) and hardening temperatures were not statistically significant in terms of survival (*F*_6,42_ = 2.042, *p* = 0.093). Overall, mean survival of *L. sativae* pupae could be ranked as follows: 5 d > 7 d > 2 d; an exception was pupae cold hardened at 3 °C where 2 d survival was greater than 7 d. For *L. trifolii,* 7-day-old pupae showed a higher mean survival than 2 and 5-day-old pupae at all hardening temperatures except at 5 °C, where the survival of 5-day-old pupae (67%) was slightly higher than 7-day-old pupae (65%). Despite the similarity of pupae of these two species, adults (both species) showed an obvious difference in RCH (Figure 2D). Survival of *L. trifolii* adults was significantly enhanced at all hardening temperatures, especially at 5 °C, where survival increased to 63% (*F*_6,14_ = 69.4, *p* < 0.001). However, after 4 h hardening, the survival of *L. sativae* adults at all critical temperatures remained 20% or less (*F*_6,14_ = 0.54, *p* = 0.76). It was found that interaction of the adults (both species) and hardening temperatures showed significant differences (*F*_6,42_ = 8.5, *p* < 0.001).

### 3.3. Durabilility of RCH

The durability of the RCH response was tested by maintaining RCH-exposed pupae (acclimated at 1 °C for 4 h) for 0, 30 min, 1 h, 2 h and 4 h at the rearing temperature (26 °C) before exposure to the critical temperature. With these treatments, the survival of both species differed significantly (*L. trifolii*: *F*_5,12_ = 116.57, *p* < 0.001, *L. sativae*: *F*_5,12_ = 109.5, *p* < 0.001; Figure 3). For *L. trifolii*, the highest survival was recorded at 0 h recovery (83.8%); however, survival of *L. trifolii* decreased as the recovery period was extended. The lowest survival was recorded at 4 h exposure (22.6%), while no significant differences were found between 30 min and 1 h recovery time. Similarly, *L. sativae* survival decreased as the recovery time was extended. The highest survival was recorded for *L. sativae* at 0 h recovery (86%), whereas survival decreased to 62% at 30 min; no significant difference was found between 30 min, 1 h and 2 h recovery times. The lowest survival rates were recorded when *L. sativae* pupae were exposed to a 4 h recovery (50%). Furthermore, survival rates of both species were compared with the recovery time. It was found that the survival of both species behaved significantly differently (*F*_5,36_ = 17.018, *p* < 0.001). Therefore, it can be observed that the recovery time has a different effect on the survival of species.

### 3.4. Determination of Supercooling Points

Supercooling points of 2-, 5- and 7-day-old pupae were measured. SCPs were not significantly different among *L. trifolii* pupae of different ages (*F*_2,87_ = 2.046 *p* = 0.135) (Table 1). The lowest SCP in *L. trifolii* pupae was recorded for 5-day-old pupae (19.5 °C). Similarly, *L. sativae* pupae of different ages showed no significant differences in SCPs (*F*_2,87_ = 2.085, *p* = 0.130) (Table 1). The lowest SCPs were recorded in 5-day-old pupae (−10.6 °C) followed by 7-day-old pupae (−9.3 °C). After further two-way analysis of SCPs of species compared with age of pupae, it was found that there was no significant difference in their SCP (*F*_2,180_ = 0.789 *p* = 0.456). SCPs of the no RCH and RCH group of both species are shown in Figure 4. For this experiment, 7-day-old pupae for *L. trifolii* and 5-day-old for *L. sativae* were selected. *L*. *trifolii* (7-day-old) and *L. sativae* (5-day-old) pupae cold harden at 1 °C resulted in the highest survival among all hardening temperatures. The no RCH group was taken from 26 °C, while the RCH groups were first treated at 1 °C for 4 h then SCPs were measured. There were significant differences in SCPs for both species in the no RCH and RCH treatments (*L. trifolii*: *t* = 2.051, *p* = 0.045; *L. sativae*: *t* = 2.602, *p* = 0.012). Mean SCPs of the no RCH and RCH treatments were 19.1 °C and 19.6 °C for *L*. *trifolii* and −10.65 and −12.48 for *L. sativae,* respectively. SCPs of both species decreased after RCH treatment.

## 4. Discussion

Prior research has focused on seasonal adaptations to cold stress, especially with respect to diapause, an environmentally programmed period of dormancy [36,37]. Since the discovery of RCH in 1987, studies on this phenomenon have escalated, especially in *Drosophila melanogaster* [38,39,40,41,42]. Both prior reports and our current results indicate that *Liriomyza* spp. differ in cold tolerance and SCPs. Leafminers are freeze avoiding, and they can tolerate sub-zero temperatures by supercooling [34]. However, no prior research has been conducted to compare RCH in *L. trifolii* and *L. sativae*. Therefore, this study compares physiological adjustments in *L*. *trifolii* and *L. sativae*, e.g., cold tolerance, RCH, and the effects of RCH on survival and supercooling. Survival of the two species decreased significantly when exposed to a gradient of cold temperatures, and *L*. *trifolii* pupae were more cold tolerant than *L. sativae* pupae. Interestingly, adults of both species exhibited similar levels of cold tolerance; in this respect our results are similar to prior studies where thermal resistance of *L*. *trifolii* was stronger than *L. sativae* [14,43]. 

RCH is widely used by ectotherms to cope with sudden changes in temperature and has been documented in many insects at temperatures of 0–5 °C [35,38,44,45]. In arthropods, RCH can be induced by different hardening temperatures and is one of the quickest acclimations to thermal variability. Variations in daily temperature have increased over the last 40 years [46]; therefore, studies of plastic responses such as RCH will contribute to efforts to predict the effects of climate change on ectotherms [47]. In the present study, RCH was induced in pupae of different ages at all hardening temperatures (0–5 °C) of both species. Acclimation of *L. sativae* pupae significantly increased survival in laboratory and field populations; however, field populations had a more active response to cold acclimation [15]. Initially, RCH was presumed to only be useful in laboratory studies but subsequent work has clearly demonstrated its ecological relevance. In *D. melanogaster,* cooling by 0.1 or 0.05 °C/min promoted RCH and enhanced cold shock tolerance [48]. In a natural environment, night time is cooler than day time. For example, in Michigan (USA), *D. melanogaster* experiences diurnal cooling from ~22 °C to ~10 °C at a rate of 1.3 ± 0.1 °C h^−1^ [49]. Flies removed at the coolest time of the day were more cold tolerant than those tested at the beginning of the cooling period [49]. A similar field induction of RCH has been reported in *D. melanogaster* [50] and in *Bactrocera oleae* [51]. These findings suggest that RCH enables insects to track temperature fluctuations and optimize cold tolerance in real time. Our results suggest that RCH did not occur in *L. sativae* adults; in contrast, RCH was induced in *L*. *trifolii* adults at all hardening temperatures. These findings warrant further study because some tropical species are capable of RCH [39]; however, the phenomenon is not universal [52,53]. By comparing RCH response in both species, it can be concluded that the survival of pupae of different ages of *L. sativae* is comparatively greater than that of *L*. *trifolii*. 

RCH-mediated protection from cold shock decreased significantly for pupae of both species as recovery time increased. Survival was highest at 0 h recovery time followed by 30 min recovery; *L*. *trifolii* protection disappeared at 4 h, whereas *L. sativae* survival was about 50%. In *D. melanogaster* exposed to simulated diurnal regimes, the protection gained through RCH was lost during the warming phase [54], and this observation is consistent with other studies demonstrating the plastic nature of RCH [55,56]. Other studies reported that survival decreased to around 20% when RCH-exposed insects were maintained for 1 h at room temperature [57]. An identical phenomenon was observed in *Frankliniella occidentalis* [35], *Musca domestica* [28], *Corythucha ciliate* [58] and *Psacothea hilaris* [59]; meanwhile, in *Euseius finlandicus* and *Bactrocera oleae* [44] RCH lasted for 2 h and 0.25 h, respectively [44,51]. Our results suggest that RCH was observed for over 2 h in pupae of both species. 

Many research studies have focused on the measurement of SCP because this provides an anchor point about which the cold tolerance strategy can be determined [22]. Similarly, SCP was previously reported as an index of cold tolerance for leafminer pupae [60]. In the present study, the SCP of *L*. *trifolii* was significantly lower than *L. sativae*. Previous studies also confirmed that the SCP of *L. trifolii* pupae is much lower than *L. sativae**,* indicating a huge difference in SCPs of the two species [14,26,28]. As described by Bale (1996) [61], there are five classes of insect cold hardness, and based on our survival data and SCP data, *L. trifolii* is chill susceptible and *L. sativae* is freeze avoiding. One of the differences between insects of these two classes is whether SCP is a reliable indicator of cold hardness. In this study, there was statistical difference between the SCPs of the RCH and no RCH groups of both species. As a freeze avoidance species, SCP is a reliable estimate of cold hardness of *L. Sativae* and the difference recorded in *L. sativae* is 2 °C, which was greater than the difference in *L*. *trifolii*. One of the major ways organisms protect themselves from sudden changes in their environment is through the phenotypic plasticity of physiological performance and tolerance of extremes [62]. Acclimation to low temperatures can improve the supercooling ability of insects [63,64]. Additionally, changes in the SCP with treatment or season can indicate biochemical or physiological changes, which may help increase survival [65]. The different cold hardness strategies of *L. trifolii* and *L. Sativae* may affect their reaction to RCH treatment and the underlying mechanism remains to be discovered.

Generally speaking, *L*. *trifolii* showed stronger resistance to cold stress than *L. sativae* in our studies. While the RCH response of *L. sativae* pupae was more robust than *L*. *trifolii, L. sativae* adults completely failed to elicit the RCH response at all hardening temperatures. RCH treatment led to a greater increase in cold tolerance and a greater decrease in the SCP of *L. sativae*; furthermore, the durability of RCH was longer in *L. sativae* than *L*. *trifolii*. These results indicate that the RCH capacity of *L. sativae* is much greater than that of *L*. *trifolii*. A previous study compared RCH capacity across different species and genetically variable lines [39] and reported a negative correlation between basal cold resistance and RCH magnitude; this agrees with our results and suggests that the capacity of cold hardening may be constrained by a basal level of cold tolerance. The distribution of *Liriomyza* spp. is determined by several factors, including survival at low temperatures, overwintering ability, and adaptability to low temperatures [5,10]. Currently, *L. sativae* is the most dominant leafminer species in China, while *L*. *trifolii* occurs only in southern China. Differences in cold responses have a great impact on the distribution and abundance of the leafminer species. For example, *L. huidobrensis* is more cold resistant than *L. sativae*, due to which the distribution of *L. sativae* is relatively restricted to low altitudes [66]. Additionally, with stronger cold tolerance, *L*. *trifolii* has been predicted to have the potential to replace *L. sativae*. For instance, in December 2020, the lowest temperature in Beijing, a typical northern city in China, was −12 °C. The lowest winter temperature was −20 °C in January 2021, and there were eight days in January when the mean temperatures were lower than −8 °C (China meteorological data service center) [67]. It is important to mention that *L*. *trifolii* may lose its competitive edge with respect to cold tolerance if global warming triggers warmer winter weather [14]. Furthermore, with the increasing abnormal temperature fluctuations accompanied by global warming, the greater RCH capacity of *L. sativae* may increase the survival of *L. sativae* overwinter. Thus, we assume that the replacement of *L. sativae* by *L*. *trifolii* in northern regions of China will take longer than that predicted by Chang et al. [19].

Further RCH-related studies need to be carried out in closely related species who are in competition because it may contribute to the invasion of a species to a new area and the displacement of one species by another in a particular area. *B. dorsalis*, for instance, has expanded its territory gradually from south to north in China [68,69]. *B. dorsalis* adaption to low temperatures exhibits rapid cold-hardening (RCH) through elevated production of cryoprotectants [70]. Although various studies have been published on RCH, the underlying mechanisms are often unclear [27]. In the process of invasion, insects must adapt to new environments [71,72], and exploring gene expression patterns that are associated with thermal adaptability can be helpful in understanding evolutionary variations between species [73]. Thus, we plan further transcriptomic and metabolomic studies to better understand the mechanistic basis of RCH in *Liriomyza* spp.

## 5. Conclusions

This study evaluated the cold tolerance and RCH ability of two *liriomyza* species in order to better understand the low-temperature survival as well as competition between them. In this study, cold stress treatment, SCP determination, and RCH treatment were checked. In summary, our results indicate that *L*. *trifolii* is more tolerant of cold and its SCP is significantly lower than that of *L. sativae*. RCH can be induced in the pupae of both species at a range of low temperatures (0–5 °C) and *L*. *sativae* pupae are stronger in RCH ability (lowering SCP and the long-lasting effect of RCH). Differences in RCH capacity and basal cold tolerance between these two leafminer species will eventually affect their competition in the context of global climate change. 

## Figures and Tables

**Figure 1 insects-12-01041-f001:**
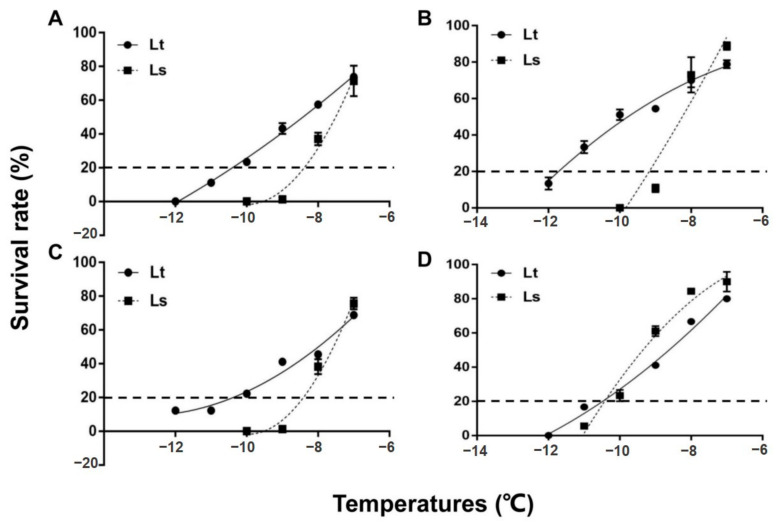
Survival of *L. trifolii* and *L. sativae* pupae and adults for determination of critical temperatures. Pupae and adults were exposed to subzero temperatures for 2 h. Panels: (**A**) survival of 2-d-old (CT: Lt, −10.4 °C, Ls, −8.4 °C), (**B**) 5-d-old (CT: Lt, −11.6 °C, Ls, −8.9 °C), and (**C**) 7 d-old pupae (CT: Lt, −10.6 °C, Ls, −8.4 °C). (**D**) (CT: Lt, −10.5 °C, Ls, −10.4 °C) survival of adults. Abbreviations: LT, *L. trifolii*; LS, *L. sativae*; and CT, critical temperature.

**Figure 2 insects-12-01041-f002:**
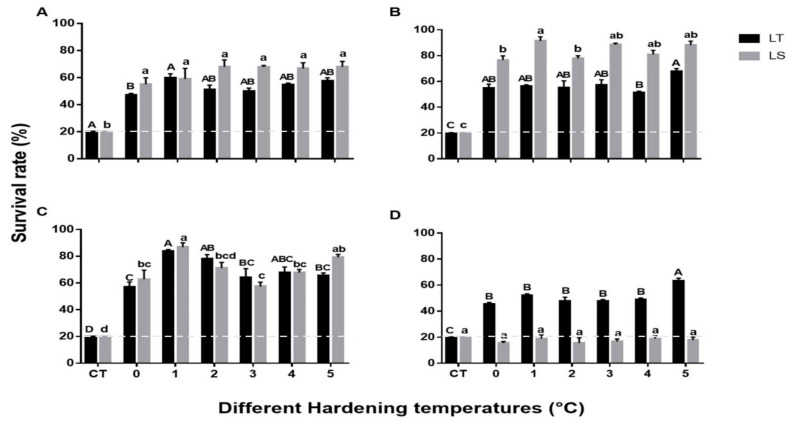
Mean survival of pupae and adults of *L. trifolii* and *L. sativae* at different hardening temperatures. Pupae and adults were exposed to 0, 1, 2, 3, 4, and 5 °C for 4 h and then transferred to the critical temperature for each life stage (shown in caption of Figure 1) for 2 h. Survival was examined after pupal emergence while adult survival was checked after a 20 min recovery period. Different letters indicate significant differences; lowercase and uppercase letters were used for *L. sativae* and *L. trifolii*, respectively. Abbreviations: LT, *L. trifolii*; LS, *L. sativae*; and CT, critical temperature. Panels: (**A**) 2-d-old pupae, (**B**) 5-d-old pupae, (**C**) 7 d-old pupae, and (**D**) adults.

**Figure 3 insects-12-01041-f003:**
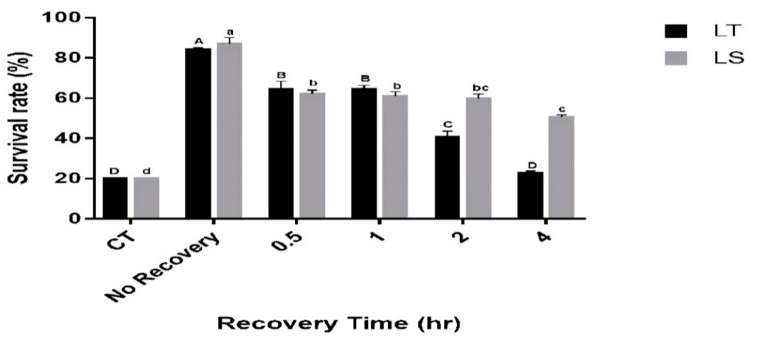
Mean survival of *L. trifolii* and *L. sativae* pupae in experiments for RCH durability. RCH-exposed pupae were either transferred directly (no recovery) or incubated for 0.5, 1, 2 or 4 h at 26 °C and then transferred to the critical temperature for 2 h (CT: Lt, −10.6 °C, Ls, −8.9 °C). Survival was determined after pupal emergence. Different letters indicate significant differences. Uppercase and lowercase letters represent *L. trifolii* and *L. sativae*, respectively. Abbreviations: LT, *L. trifolii*; LS, *L. sativae*; and CT, critical temperature.

**Figure 4 insects-12-01041-f004:**
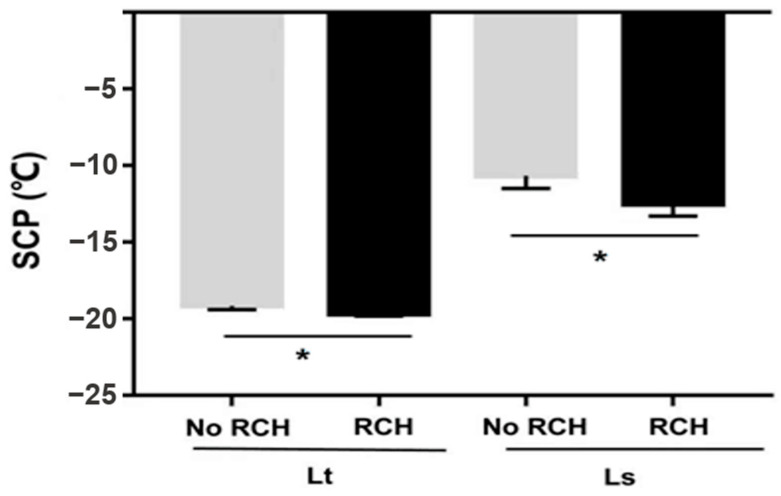
Mean supercooling points (SCP) of *L. trifolii* and *L. sativae* pupae in experiments to evaluate the impact of RCH on SCP. Data were analyzed by the Student’s *t*-test, *p* < 0.05. Asterisks represent significant differences between RCH and noRCH.

**Table 1 insects-12-01041-t001:** Mean supercooling points (SCPs) ± S.E of pupae of different ages.

Age of Pupae (Days)	*L. sativae* (°C)	*L*. *trifolii* (°C)
2	−8.67 ± 0.61	−19.1 ± 0.14
5	−10.65 ± 0.84	−19.5 ± 0.1
7	−9.30 ± 0.61	−19.2 ± 0.25
Statistical analysis	*F*_2,87_ = 2.085 *p* = 0.130	*F*_2,87_ = 2.046 *p* = 0.135

## Data Availability

Data are contained within the article.

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
