# Peer review of "Differential Response of Leafminer Flies Liriomyza trifolii (Burgess) and Liriomyza sativae (Blanchard) to Rapid Cold Hardening"

_insects, 2021, doi:10.3390/insects12111041_

Round 1
Reviewer 1 Report
Liriomyza trifolii (Burgess) and Liriomyza sativae (Blanchard) are closely related species and are important polyphagous leaf-mining pests worldwide. In this manuscript, the authors investigate the the cold tolerance of these two leafminer species under RCH. This study is interesting and reveal the potential mechanism of RCH and interspecific competition between L. trifolii and L. sativae. The manuscript is well written and organized, however, it requires some minor improvements. I recommend to accept the manuscript after the following comments are addressed:
- In Figure 1, the author should write the specific nonlinear model in detail, and 80% mortality threshold line for critical temperature should be added in the figure.
- Some errors in the text need to check and modify. In Line 162 and 165, P should be italicized; In Line 180, L. trifolii and L. sativae also needs italics...
- The CT group appears abrupt in Figure 2 and 3. It is suggested that the authors express it in another form or remove it directly. In Figure 2, please explain what the ABCD represents.
- The authors argue that, In Line278-279, “In this study, pupae of L. trifolii comparatively more cold tolerant than pupae of L. sativae". However, in figure 2, it seems that L. sativae has stronger RCH ability than L. trifolii. why?
- About this manuscript, the method is correct and the result and data are reasonable. But, as far as we know, temperature may not be the limiting factor in the distribution of L. trifolii and L. sativae. Besides environmental factors, interspecific replacement of the two species were determined by their direct competition on host in some cases. Avoid over discussing interspecific competition based only on low temperature tolerance.
Reviewer 2 Report
I have read the manuscript entitled “Response difference of rapid cold hardening in two leafminer flies, Liriomyza trifolii (Burgess) and Liriomyza sativae (Blanchard)” submitted to Insects. The authors compare the cold tolerance (supercooling point) of two related invasive species in China (L. trifolii and L. sativae). They did so in several sub-zero temperatures and several pupa ages and adults. They also examined the effect of rapid cold hardening on cold tolerance (survival rate and supercooling point). They detected a complex pattern: differences between the two species, and perhaps an interaction of species and sub-zero temperature (in the first experiment).
I liked the study and think the authors did a thorough work. I think that the data deserve to be published. This study adds to our knowledge of thermal tolerance of species, which is important to agriculture. It may also explain why one species is a more successful invader than the other. The methods and concepts are not novel, as many other studies focus on cold tolerance of insects. However, this is an elegant study, examining two species under several temperatures, several life stages, and both cold tolerance and the contribution of cold hardening. Still, I think there are 3 points for improvement: (1) The most important point relates to the statistics. I did not understand the statistics conducted and think it is suboptimal. If you really wish to compare between species and life stages, the best would be to include all explanatory variables in the analysis. In other words, if you examine survival, then this should be the response variable, and you should include 3 explanatory variables as well as all possible interactions: species, life stage, and temperature. (2) I find some parts of the manuscript to be too wordy. While this is not a major point, shortening it by removing repetitions will contribute to the readability of the manuscript. (3) There are numerous errors in English. Although I am not a native speaker, some sentences are clearly wrong. I recommend the author to have another look. I also recommend on the free software Grammarly, which can detect many of such errors. In general, however, it is an interesting study that can be accepted after a major revision. Please see specific comments below.
Line 46: Please add the order and family of the studied insects.
Line 48: coma instead of full stop after [1].
Line 49: All America? North America? South America?
Line 56: species instead of specie.
Lines 56-57: The sentence is unclear. Do you mean that L. trifolii has been after its invasion a more damaging pest of crops in China than L. sativae?
Lines 58-59: Are you sure that high cold tolerance is the only reason why one species is displacing the other species? This may hold for northern China, but what is your explanation for other regions in China? Furthermore, how do these insects survive winter? Do they hibernate and if so at which developmental stage?
Lines 66-67: This is true mostly in high latitudes. I think it will be good to mention also other factors affecting distribution ranges, except for temperature. There are plausibly many other factors involved.
Line 68: Explain for non-specialists what the supercooling point is.
Line 71: Replace “Recently due to increasing of global warming” with just “owing to global warming”.
Line 72: Add a full stop after “to increase further”. Also, why is it important to study cold tolerance if we experience global warming? I would say then that studying heat tolerance is much more important. Please explain as this is not trivial.
Lines 73-76: Writing a few words on the mechanisms of each acclimation type will make the distinction between them clearer to the reader.
Line 83: Please change to “RCH does not only increase…”.
Line 84: Unclear what “ecological functions” mean.
Lines 85-91: This is a repetition of lines 71-73. Please bring this information up and merge the sentences together. It is enough to discuss once the implications of cold tolerance to global warming, but please explain because it is not trivial. Try not to have such repetitions, because it elongates the manuscript without a good reason.
Line 94: Remove “phenomenon”, it is superfluous.
Line 95: I agree that cold tolerance may play a role in the competition between the two studies species. However, many other factors may play a role as well. This should be made clear earlier.
Lines 95-96: Can you please add more specific predictions after your hypothesis. How should the RCH affect the competition?
Line 106: add “the” before “procedure”. Lines 110-111: add “the” before “field” and “experimental”.
Lines 113-114: Maybe change to: “We examined the critical temperature, which causes 80% mortality (lethal temperature 80) by exposing pupae of 2, 5, and 7 days and adults of 2 days to the following sub-zero temperatures: -7, -8, -9, -10, -11 and -12°C for 2 hours. Each thermal treatment consists of 30 individuals per stage”. Also, do you mean you examined 30 pupae per temperature for each of the mentioned ages (2, 5, and 7)? Does it mean that the total number of adults is 6x30 = 180, and the number of pupae is 3x6x30 = 540?
Line 119: Is temperature controller = climate cabinet? Also, were light and humidity controlled or monitored?
Line 120: 26°C under light/dark conditions? What was the humidity?
Line 122: What do you mean by “Three replicates”? Each treatment already comprised 30 individuals, right? So you repeated it for 3 times?
Line 131: How did you decide on the time of exposure (4 hours)?
Lines 133-134 (Different stages of insects…) seem to be a repetition of lines 131-132.
Line 137: acclimation temperature or hardening temperature? Please use consistent terminology.
Line 139: “To determine the duration of RCH response pupae both the species” – the word pupae is unclear. You mean perhaps “of the pupae”?
Lines 157-159: I don’t get how the data were analyzed. What is the response variable? What are the explanatory variables? If you use non-linear regression, which function did you use? I would analyze the data differently: Why not just using linear regression with survival as the response variable (here I think you need an arcsin transformation), and age, temperature, and species as three explanatory variables. You should check for all two-way interactions and the single three-way interaction species x age x temperature.
Line 159: Add – “followed by” before “Tukey’s multiple comparison”.
Lines 162, 164: No need to write “differences were considered statistically significant when P<0.05”. It is clear.
Line 167: temperature instead of Temperature. Line 169: Species names in italics (please check the rest of the manuscript).
Lines 168-170: This belongs to the Methods.
Lines 174-178: This is an interesting result. I wonder whether such temperatures are common in northern China and how often? Maybe adding this in the Discussion will be of interest, in order to realize how this finding fits the natural environment.
Lines 194-197: This belongs to the Methods.
Line 201: of both species. Also here, I would use fewer tests with more variables examined in each test. Then, you can more easily compare between the species.
Figure 2: Write in the figure captions what the panels a, b, c, and d mean, liked you did in Figure 1.
Lines 228-231: I think that such “introduction” to each part of the experiment is not really needed. Instead, here, maybe add a sub-heading. Also in this experiment, in order to better compare among species, ages, and temperatures, I would use a single test with all three factors included as explanatory variables.
Lines 247: Determination of supercooling point: Also here, the statistical comparison between species is of interest.
First paragraph of the Discussion: Maybe elaborate here a bit also the results not related to RCH. For example, the supercooling point, differing between the two species. This result (presented in figure 1) is interesting on its own right.
Line 277: You have “decreased” twice here.
Lines 278-279: Any comparison between the two species will be better conducted if you include the species as a variable in the statistics. You will be able to support your conclusions better.
Line 320: Here you might want to mention Andersen et al. (2015 Functional Ecology 29:55-65), who suggested that SCP is the best measure of cold tolerance, at least in Drosophila melanogaster.
Lines 364-372: I find this part a bit lengthy and not very related to your work. I would shorten it. Similarly, I would remove lines 384-386: it is not a part of your research.
Line 364: If you mention the RCH mechanisms, then maybe mention what they are in the Introduction.
Discussion in general: I find the Discussion to be a bit wordy with jumps between topics. Maybe use sub-headings and think again how not to repeat the same sentence more than once.
Reviewer 3 Report
General comments:
This study exposed two species of leafminer flies to acute cold exposures to identify the lethal temperatures (LT80). The introduction provides some good background material, but could use a little more information regarding the intensity and importance of the competition between the two species. Furthermore, one point about both species that could be briefly discussed is what season the pupae exist in. Are either species exposed to winter as pupae and/or adults? Knowing this upfront could provide relevance (or question) your testing of different life stages. The methods are fairly good, but could use a few more details (see specific comment below). One concern is the lack of clarity of sample sizes. This should be an easy fix, but the details are missing (i.e., total n and n per age and treatments). The results are straight forward, though I have a few minor questions regarding the figures (see below). This discussion does a good job interpreting the results and incorporates the literature well. Similar to my comments about the introduction, a little more focus could discuss the actual competition between the two species and the relevance of testing different life stages for cold tolerance.
Specific comments:
Line 48: replace the period with a comma.
Line 49: by “America”, do you mean both North and South Americas? If so, change to “the Americas” or specify which “America” you are referring to.
Lines 82-82: “RCH has been observed in variety of insects to increase survival to lethal cold stress [25,28]”. I’m not sure this means what you intend it to mean. How does an organism survive a lethal cold stress? Are you saying that RCH improves cold survival at temperatures previously lethal without the onset of RCH? Please clarify.
Line 118: I assume the “n=30” for pupae is really n=90? That is, were 30 different pupae tested at each age (2, 5, and 7-days)?
Lines 143-155: while I follow your methods of SCP, a citation showing where the methods came from would be helpful, and confirm that these are standard methods.
Lines 187-188 (figure 1): It seems some data points do not have error bars. Unless those groups had zero variance? Also, explain what the error bars represent (i.e., standard deviations, standard errors, confidence intervals).
Lines 220-221: why are capital and lower case letters mixed? For example, is “A” different than “a”? Please fix or clarify the use of mixing upper and lower case letters.
Round 2
Reviewer 2 Report
I have read the revised manuscript, which has been greatly improved. The vast majority of my comments were accepted. The authors did a good job, in my opinion. Here are some more minor comments:
Line 19: Remove “always”.
Lines 179, 180: species instead of specie.
Lines 230-231: “It was found that interaction of 5 days old pupae of both the species differs significantly in their survival rates (F6,42=8.5, p<0.05)” – I am not sure I get this sentence. Interaction between which two factors? Species and what? Same for lines 234-235 – the intention is unclear.
All over the results: Please provide exact P values and not only P < 0.05 or P > 0.05.
Lines 244-245: The phrasing is unclear. Interaction of which two factors?
Line 270: Duration of time – where? Please make it clearer without the need to look again at the methods.
Lines 286-288: The sentence is not grammatically correct. Please check the sentences added during the revision because many of them are hard to understand and are not in good English.
